# Exploring Hygiene Behaviours Among Child Caregivers in Rural Malawi Using Multilevel Logistic Models

**DOI:** 10.3390/ijerph22050801

**Published:** 2025-05-20

**Authors:** Collins Anusa, Salule Joseph Masangwi, Kondwani Chidziwisano, Tracy Morse

**Affiliations:** 1Department of Mathematics, Malawi University of Business and Applied Sciences, Blantyre 303, Malawi; 2Centre for Water, Sanitation, Hygiene and Appropriate Technology Development, Malawi University of Business and Applied Sciences, Blantyre 303, Malawi; smasangwi@mubas.ac.mw (S.J.M.); kchidziwisano@mubas.ac.mw (K.C.); 3Centre for Sustainable Development, University of Strathclyde, Glasgow G1 1XJ, UK; tracy.thomson@strath.ac.uk

**Keywords:** child caregivers, multilevel binary logistic models, hygiene, behaviour

## Abstract

This study aimed to explore the factors influencing food hygiene behaviours among child caregivers in Chikwawa district, Malawi. This research focused on three specific hygiene behaviours: keeping utensils on an elevated surface, using soap to clean kitchen utensils, and washing hands with soap at critical times. These practises are known to contribute to the reduction in diarrhoeal disease. To understand these behaviours, this study utilised multilevel binary logistic models to examine variations at both the household and village levels. The findings reveal that educational background, age group, occupation, self-confidence, intervention, self-will, and perception were the most significant factors influencing food hygiene behaviours. Notably, there were significant variations at the village level (*p* < 0.00001), while no significant variations were observed at the household level (*p* > 0.1). Additionally, caregivers from areas where interventions were implemented showed a positive response to these interventions.

## 1. Introduction

As a way of reducing diarrhoea cases among children, the World Health Organization (WHO) has recommended important parameters that need to be followed at the household level [1]. These parameters include access to safe water, improved sanitation facilities, exclusive breast feeding, vaccination, access to vitamin A, hygienic weaning practises, and improved personal and household hygiene. There are high chances that when food is prepared under unhygienic conditions, it can become microbiologically contaminated [2]. Studies have shown that utensils, such as spoons, cups, pots, baby bottles, plates, etc., are potential vectors of pathogens such as *Escherichia coli*, *Salmonella*, and *Vibrio cholera* [3,4]. The contamination of utensils is usually associated with the method of cleaning, resulting from repeatedly using the same water, such as water that was used to wash dirty clothes [5]. To avoid contamination, utensils have to be properly cleaned and stored [6]). Other studies conducted by various researchers have shown that handwashing with soap alone can reduce diarrhoea incidence by 30–47% [7,8]. Since the use of soap in washing hands and kitchen utensils has proven to effectively contribute to enteric pathogen reduction, it is important to understand the characteristics that affect an individual’s psychological and social factors that drive handwashing practises and the context in which they occur. Changing an individual’s behaviour is a process that requires change in contextual and psychosocial factors that predict human behaviour in a given setting, such as attitudes, norms, and self-regulation attributes [9]. The said contextual factors are the environment in which the behaviour occurs, and they include personal (e.g., age and literacy), social (e.g., economic conditions), and physical parameters (e.g., the presence of sanitation facilities such as a handwashing facility), whereas psychosocial factors have been defined as the influence of social factors on an individual’s mind or behaviour, as well as the interrelation of behavioural and social factors [10]. Caregivers who have access to hand washing facilities are more likely to wash their hands at critical times [6]. Caregivers who keep domestic animals are more likely to keep utensils on an elevated surface. The use of soap for hand washing is uncommon among caregivers from low-income households because they prioritise it for other household usages, such as washing clothes and bathing [6]. Caregivers who have limited access to water facilities tend to repeatedly use the same washing water, which results in the contamination of utensils [6]. However, there are barriers to hygiene practises among child caregivers, especially in low-income countries. These include a lack of resources (such as soap and water) and an enabling environment, including monetary decision making power and social support [11]. Labour-saving hygiene technologies that allow for utilisation by both men and women, for example, and improving household water resources and raising awareness about hygiene for both men and women can increase hygiene practises in households [12,13]. Social–cultural norms also hinder optimal hygiene practises among child caregivers [11]. Some household members are bound by approval from senior people in the household to use certain hygiene resources for home use [6].

There are several interventions that have focused on improving hygiene practises among child caregivers in Malawi and other low-income countries. These include the involvement of community health volunteers (already active in the area) in delivering household-level food hygiene interventions, who promote key hygiene behaviours such as hand washing with soap at critical times and using soap to clean kitchen utensils [3,6,14].

Multilevel logistic models are regression models that allow the analysis to take into account the hierarchical nature of the data, to investigate sources of variations within and between hierarchies, as a way to investigate which parameters predict variations and to describe which parameters predict hierarchical differences. This implies that using a single-level ordinary regression for such hierarchical data may result in biassed standard errors for parameter estimates, which might result in incorrect inference [15]. There have been studies in Malawi which have shown that both individual- and community-level characteristics are important considerations for policy makers in designing interventions [16,17].

This study used mid-term evaluation data from the Sanitation and Hygiene Applied Research for Equity (SHARE) project, which was implemented by the Centre for Water, Sanitation, Hygiene and Appropriate Technology Development (WASHTED) of the Malawi University of Business and Applied Sciences, to explore household and village effects using multilevel binary logistic models. The study mainly focused on variations between areas of intervention as the treatment group and areas where an intervention was not carried out as the control. The project delivered various interventions which were in different categories of treatments in respective villages. However, this study concentrated on the overall impact of all of the interventions as they were all delivered for the same purpose of trying to change caregivers’ hygiene behaviours. The distinction was made by forming a binary variable which provided a score of 1 for treatment and 0 for the control. And by looking at the intervention variable in the regression model, it was able to be seen whether there were differences between the control and the treatment. This was achieved by assessing variations amongst child caregivers in relation to washing hands with soap at critical times, storing kitchen utensils on a raised surface, and using soap to clean kitchen utensils. If not performed properly, these behaviours are said to be highly associated with the spread of diarrhoea, especially in developing countries [1].

The assumption of this study was that behaviours within villages and households are related, and that there is no relationship regarding what happens between villages and households, even in cases where they are adjacent to each other. So, using a single-level ordinary regression for such hierarchical data may result in biassed standard errors for the parameter estimates [15].

The objectives of this study were, therefore, to use multilevel binary logic regression models to explore the following:Whether there are variations in hygiene behaviours among child caregivers between households in relation to the area of intervention and the non-intervention area;Whether there are variations in hygiene behaviours among child caregivers between villages in relation to the area of intervention and the non-intervention area.

### 1.1. Study Rationale

This study helped us to understand that conducting an evaluation using ordinary regression for hierarchical data may result in biassed standard errors for parameter estimates, which might result in incorrect inferences. It was important to conduct this study to uncover factors that lead to variation at different levels, while indicating whether interventions have an impact or not. The application of models of hygiene behaviours can help us understand factors that affect hygiene interventions among child caregivers in rural communities, and this will further help WASH experts to design effective, context-specific hygiene interventions to promote hygiene behaviours in rural communities.

### 1.2. The Study’s Conceptual Framework

This study applied the RANAS model, which provided guidance in the identification of the behavioural factors. The model further provided an understanding that human behaviour occurs in an environmental setting, where a number of factors come into play. In this case, the model provided guidance on details of contextual factors included in the study questionnaire. A detailed explanation of the RANAS model is provided in previous publications [3,6].

## 2. Methods

### 2.1. Study Population and Setting: Sampling and Data Collection Methods

A survey was conducted in Chikwawa district in Malawi, Southern Africa, and comprised a sample of households within villages from Traditional Authorities (TAs). According to the data collected in the 2018 population census, the number of people living in the district was estimated at around 564,684 [18]. The survey was conducted in four rural administrative TAs: Masache, Ngowe, and Ngabu as the areas of intervention and Maseya as the control. A total of 850 households nested within 46 randomly sampled villages were selected. Each village had at least 20 respondents. The inclusion criteria for a household to be part of the study were that it should be located in either the intervention or control area, have a functioning latrine, and resides within a 500 m radius of a functioning borehole to ensure that there are no significant variations in access to water or sanitation infrastructure. In addition, eligible households needed to have a child aged between 4 and 90 weeks at the time of enrolment to ensure that the children were not neonates and that all children would be younger than 60 months at the end of the intervention period. The ages of the children were verified, where possible, through birth and/or immunisation records supplied by the caregiver and cross-checked by community health workers (health surveillance assistants [HSAs]). The main caregiver of the child was selected as a study participant from each household. Details of the sampling process, data collection methods, and the selection criteria of the TAs are provided in previous publications [3,6].

### 2.2. Definitions of Key Concepts

Child caregiver: Any household member, including parents, who are responsible for the daily care of young children.Multilevel binary logistic: A type of regression analysis used when the dependent variable is binary, meaning it has two categories. It is commonly used when the outcome is coded as “1” or “0”.Behaviour: The performance of a particular action. This includes the execution of both healthy and unhealthy behaviours.Hygiene: Conditions and practises that serve to promote or preserve health in the household.

### 2.3. Variables

#### 2.3.1. Outcome Variables

The outcome variables in this study were the following:IKeeping utensils on an elevated surface;IIUsing soap to clean kitchen utensils;IIIWashing hands with soap at critical times.

##### Questions on Targeted Behaviours

As indicated in Table 1 below, responses to the questions from the questionnaire were recorded on a 5-point Likert scale (ranging from “not at all” to “very much”, where 1 denotes not at all, 2 denotes little, 3 denotes moderate, 4 denotes much, and 5 denotes very much). The variable “washing hands with soap at critical times” was a product of the transformation of the variables on washing hands with soap before and after eating, washing hands with soap after vising the toilet, washing hands with soap before food preparation, and washing hands with soap after removing a baby’s nappy. This was achieved by taking an average of all responses such that those that were at or below the average value were considered those that were rarely washing hands with soap at critical times. Those that were above average were considered those that were mostly washing hands at critical times.

In order to transform outcome variables to binary variables, all factors falling at or below the mid 3-point value on a scale of 1–5 were considered non-doers of the targeted behaviours and were recorded as 1, whereas factors at or above 4 were considered doers of the behaviour and were recorded as 0. Behavioural outcome was modelled to allow an examination of independent variables as determinants for doers and non-doers.

#### 2.3.2. Explanatory Variables

##### Description of Explanatory Variables

Table 2 shows description of explanatory variables included in the model. The variables were obtained by first checking the simple relation between each variable factor and the outcome variable of interest while ignoring all other variables. In this regard, only variables that were significant at *p* ≤ 0.05 with a Deviance Information Criterion (DIC) reduction of at least 5 were selected for the final model. Fifteen variables satisfied this requirement: educational background, gender, the presence of faeces outside the house, age group, marital status, occupation, intervention, relative wealth, health knowledge, risk, effort, norm, confidence, willingness, and perception of price.

Wealth index was derived using the method of “variations” that assigns weights to indicator variables and uses the inverse of the proportion of the number of households with an asset or service as the weight for the indicator [19]. A categorical variable was then derived by cutting the wealth index distribution into three distinct segments to give low-, middle-, and high-wealth categories.

Health knowledge, risk, effort, norm, confidence, willingness, and perception of price were derived by computing the weighted average of all relevant variables. For example, the health knowledge variable, which had two responses (Yes = 1; No = 0) was derived by adding all responses for variables relating to health knowledge on the spread of diarrheal disease. The sum was then divided by the total number of related variables (4). If the result was less than or equal to two, a score of 0 was assigned, and a score of 1 was assigned if the result was greater than two.

#### 2.3.3. Analysis and Estimation

The data analysed in this study have two levels: the household and village levels. In order to quantify variations in child caregivers’ behaviours between villages, multilevel modelling was used to analyse the data with households as level 1 and villages as level 2. A series of two-level binary logistic regression models were constructed to test their patterns of variation and corresponding risk factors, and any action taken when a member of a household was considered a non-doer was used as a response variable.

The binary regression model [20] was used to explain the probability of the outcome as a function of independent variables. If the respondent representing ith household from jth village reported to be a non-doer in their household on the day of the survey, then a response was written as follows:$$y_{ij}=\left\{\begin{array}{c} 1\\ 0 \end{array}\right.\begin{array}{c} if\;ith\;household\;is\;a\;non-doer\\ Doer \end{array}$$
where $y_{ij}\left|\pi_{ij}=Ber\left(\pi_{ij}\right)\right.$, and logit $\left(\pi_{ij}\right)=x_{ij}\beta +u_{0j}$ is a random component model.

$i=1,\ldots ,i_{j}$ households and j=1,…,j villages, with $\pi_{ij}\;$ being the probability that the ith household in the jth village is a non-doer. Vector β is a regression coefficient corresponding to covariate $x_{ij}$.

Variation at the village level is modelled through $u_{0j}$ such that $u_{0j\;}-\left(\sigma_{u}^{2}\right)$. Considering the complexity and high-dimensional nature of the model being used, which can make it difficult to find solutions analytically, an estimation was performed using Bayesian procedures in MLwiN 3.05 software (University of Bristol, Bristol, UK). Stability of all parameters was monitored by observing the Raftery–Lewis diagnostics [21]. The maximum number of iterations performed to achieve stability was 5000. This model is able to detect whether there are significant differences between non-doers and doers with their corresponding determining factors.

#### 2.3.4. Ethical Consideration

The University of Malawi’s College of Medicine Research Ethics Committee (P.04/16/1935) approved the study protocol. Permission was obtained from the Chikwawa district council, Chikwawa district health office, and the traditional chiefs. The study was registered with the Pan African Clinical Trials Registry (sPACTR201703002084166).

## 3. Results

### 3.1. The Demographic Characteristics of the Study Participants

As indicated in Table 3, In terms of education, 21% of the participants never attended school, whereas ±70% went to school until the primary level and ±10% reached the secondary level and above. Most of the respondents were in the age category of 18–28 years (59%), whereas 29% were 29–38 years, 10% were 39–48 years, 1% were 49–58 years, and 1% of participants were over 58 years. In terms of occupation, 72.9% of the study population were farmers, whereas 14.9% were business women, and 12.1% were employed. Regarding marital status, 9% were single, and 91% were married. In terms of occupation, 72% of the participants were farmers, whereas 15% were in business, and 2% were employed. The study included approximately 22% of the participants as a control group, and 78% were from intervention areas. Regarding social–economic status, 86.5% of participants were from the low-wealth category, 10.8% from the middle wealth category, and 2.7% of participants were from the high-wealth category [19].

### 3.2. Descriptive Estimates of Non-Doers of Washing Hands with Soap at Critical Times, Using Soap to Clean Kitchen Utensils, and Keeping Utensils on a Raised Surface in the Area of Intervention and the Non-Intervention Area at the Household Level

Table 4 presents descriptive estimates of non-doers of washing hands with soap at critical times, using soap to clean kitchen utensils, and keeping utensils on a raised surface between the area of intervention and the non-intervention area at the household level. The results show that there was a higher percentage of caregivers from non-intervention areas who were not washing hands with soap at critical times, were not using soap to clean kitchen utensils, and were not keeping utensils on a raised surface compared to the percentage of caregivers in the intervention area who were not performing these three behaviours. For instance, 68% of caregivers in non-intervention areas were not keeping utensils on a raised surface, whilst 4% of caregivers in the intervention areas were not keeping utensils on a raised surface. There was 30% of caregivers who were not washing their hands with soap at critical times in the intervention areas, while in non-intervention areas, 3% of caregivers were not washing their hands with soap at critical times. Twenty-two percent (22%) of caregivers from intervention areas were not using soap to clean kitchen utensils, while three percent of caregivers from non-intervention areas were not using soap to clean kitchen utensils.

### 3.3. Variation in Hygiene Behaviours Between Households and Between Villages in Area of Intervention and Non-Intervention Area in Relation to Controlled Risk Factors

Table 5 presents data on a series of binary logistic regression results relating to keeping utensils on an elevated surface, using soap to clean kitchen utensils, and washing hands with soap at critical times. Educational background, the age group of caregivers, occupation, interventions, relative wealth, self-confidence, willingness to practise proper hygiene, and perceptions of price were all factors that were associated with keeping utensils on an elevated surface, using soap to clean kitchen utensils, and washing hands with soap at critical times.

Those who never attended school were less likely to wash their hands with soap than those who attended school until the secondary level and above (*p* = 0.03). Caregivers in the age group of 18–28 years were less likely to keep utensils on an elevated surface than those in the age category of 29–38 years (*p* = 0.02). Those who were practising farming were less likely to use soap to clean kitchen utensils and less likely to wash their hands with soap at critical times than those who were employed (*p* = 0.005, 0.007). Those who did not receive interventions on proper hygiene practises were less likely to keep utensils on an elevated surface, use soap to clean kitchen utensils, and wash their hands with soap at critical times than caregivers who received interventions (*p* < 0.00001).

Relative wealth was also a key factor, as it was found that caregivers in the low-wealth category were less likely to use soap to clean kitchen utensils and were less likely to wash their hands with soap at critical times than those in the high-wealth category (*p* = 0.01). Those in the low-wealth category were less likely to keep utensils on an elevated surface than those in the average wealth category (*p* = 0.08).

Caregivers who had no self-confidence were less likely to keep utensils on an elevated surface and were less likely to wash their hands with soap at critical times than those who were confident (*p* = 0.0006, 0.005). Caregivers who were not willing to use proper hygiene practises were less likely to wash their hands with soap at critical times than those who were willing (*p* = 0.06). Those who regarded proper hygiene as expensive were less likely to keep utensils on an elevated surface and were less likely to wash their hands with soap at critical times than those who thought otherwise (*p* ≤ 0.00001).

There were 30% of caregivers who were not washing their hands with soap at critical times from intervention areas, while in non-intervention areas, 3% of caregivers were not washing their hands with soap at critical times. Twenty-two percent (22%) of caregivers from intervention areas were not using soap to clean kitchen utensils, while three percent of caregivers from non-intervention areas were not using soap to clean kitchen utensils.

The results in Table 5 indicate that there were variations at the household level regarding keeping utensils on an elevated surface, using soap to clean kitchen utensils, and washing hands with soap at critical times as indicated in Figure 1, Figure 2 and Figure 3 respectively. Figure 4, Figure 5 and Figure 6 show that there there were variations between villages regarding keeping utensils on an elevated surface, using soap to clean kitchen utensils, and washing hands with soap at critical times respectively (*p* < 0.0001). This agrees with the results displayed in caterpillar plots where some villages and households are above zero on the positive side of the graph, and other households and villages are below zero on the negative side of the graph. Those that are above zero are non-doers, while those on the negative side are doers. As indicated on the legends for individual graphs, villages and households from intervention areas are shown in blue, while villages and households serving as the control are shown in red. However, it has been noted that some villages and households from non-intervention areas fall on the negative side of the caterpillar plot, implying that such villages and households were doers though they did not receive the intervention. Similarly, some villages and households from intervention areas fall above the caterpillar plot, implying that they were non-doers though they received interventions. For instance, Figure 1 shows that households from non-intervention areas were on the far negative side of the caterpillar plot. The software shows that these households are in the Kaphiri village, which is part of the control, but they are on the doers’ side, as indicated in a previous study. Factors that can lead to such outperformance may include proximity to some villages that were in the intervention area, village leadership influence, or influence form other related interventions

## 4. Discussion

The study results show that most of the participants did not finish school to at least secondary level. This could be attributed to some cultural beliefs in which parents prioritise educating boys whilst encouraging girls to get married instead. The results also indicate that most of the study participants were from poorer households. This could be connected to the fact that most of these participants were less educated, and this could mean that they were less likely to be employed in better-paying jobs. The results also reveal that most caregivers from intervention areas were doers for all three targeted behaviours compared to caregivers from non-intervention areas. This could imply that the interventions delivered had a positive influence on hygiene behaviours amongst caregivers in the intervention area.

Variation due to hierarchical effects may come as a result of unobserved factors that affect hygiene behaviours at the household and village levels. These variations may have occurred due to various contextual factors, such as access to hygiene facilities or religious affiliations of caregivers [22]. Caregivers who have access to hand washing facilities are more likely to wash their hands at critical times [6].

A number of studies analysing food hygiene behaviours among child caregivers have been carried out in Malawi [3,6]. However, the uniqueness of this study is that it employed a multilevel approach to account for village effects in addition to household factors.

The data accrued show significant variations at the household level for all targeted behaviours. However, this study has revealed a significant variation at the village level for only two behaviours: keeping utensils on a raised surface and using soap to clean kitchen utensils. The variations at the village level may have occurred due to the differences in leadership management between village heads (leaders), where some village heads may have strict, self-enforced rules and morals, while others may not care much about how their people live (i.e., some village heads may be highly civilised to enforce proper hygienic practises, while others may not care) [23]. Variations at the village level may have also occurred due to differences regarding the availability of livestock animals because livestock availability differs from one village to the other due to the availability of natural feed. This connects to the fact that most caregivers who keep domestic animals are more likely to keep utensils on an elevated surface [6]. The impact of educational background has been shown, namely in that there is a significant difference between caregivers who attended school to the secondary level and above and those who never attended school, with regard to washing hands with soap at critical times. This is expected in any society where most highly educated people are more likely to exhibit correct behaviours than those who never attended school. Age group has been shown to affect the behaviours of caregivers, whereby those aged 29–48 years were more likely to keep utensils on an elevated surface than other age categories. This might be due to the agility and maturity associated with that age group [24]. This study has shown a significant impact of occupation on using soap to clean kitchen utensils and washing hands with soap at critical times: those who were performing farming were less likely to use soap to clean kitchen utensils and less likely to wash their hands with soap at critical times than those who were employed. This might be due to exposure one has at work, where it can be expected that such workplaces have a variety of people with different backgrounds, thus increasing the chance of peer learning.

The other result of this study is the effect of interventions. The difference could be attributed to the knowledge gap in the importance of keeping utensils on an elevated surface, using soap to wash kitchen utensils, and washing hands with soap at critical times between those who did and did not receive interventions. These results are not dissimilar from [3], which studied the impact of interventions at the household level, where it was found that interventions had a positive impact on hygiene practises among households. The impact of relative wealth on keeping utensils on an elevated surface and washing hands with soap at critical times could be attributed to the fact that low-wealth households tend to have limited access to important hygiene resources like soap and clean water. This agrees with the findings from previous research, where authors argued that the use of soap for hand washing is uncommon among caregivers from low-income households because they prioritise it for other household usages, such as washing clothes and bathing [6]. Confidence in performance had an impact on child caregivers. This is likely because those with self-confidence were more likely to have enough information on the importance of being hygienic. Similarly, caregivers who were willing to wash their hands with soap had more chances to wash their hands with soap at critical times. Confidence and willingness are good measures of one’s own effort to do the right thing due to one’s own belief. This is why those who were confident and willing to exercise proper hygiene practises were more likely to keep utensils on an elevated surface, wash kitchen utensils with soap, and wash their hands with soap at critical times. The perception of price for practising proper hygiene also had an impact on washing kitchen utensils with soap and keeping them on an elevated area. This is likely because those who felt that using soap is expensive can at least afford the sacrifice to use it for washing kitchen utensils. Also, it is more likely that they would consider constructing a dish rack as costly.

The results shown in the caterpillar plots indicate that there were variations between participants from the control and intervention areas. For instance, Figure 6 shows villages above zero, indicating that caregivers in those villages were less likely to wash their hands with soap at critical times, and those below zero, a sign that caregivers from those villages were likely to wash their hands with soap at critical times. It is noted that most villages that are concentrated on the far positive side are in the non-intervention area, and those on the far negative side are from intervention areas. This shows that intervention is one of the risk factors associated with variation for the villages in this study. This implies that interventions had a positive impact on the hygiene behaviours of caregivers in Chikwawa, agreeing with previous researchers [3]. However, the credible intervals in a previous study are all overlapping, indicating that there are no significant differences between villages regarding washing hands with soap at critical times.

These findings agree with some researchers who have found that behavioural determinants, including norms, willingness, self-confidence, support mechanisms (i.e., interventions), risks, and other broader contextual factors, are likely to affect the behaviour of hand washing with soap at critical times [25].

The observations in this study show that interventions had a positive impact on hygiene behaviours amongst child caregivers. It can also be concluded that there were variations at both the household and village levels for all three hygiene behaviours.

## 5. Conclusions

This research confirms that caregivers from the area of intervention had a positive response to the interventions. The study also revealed that educational background, age group, occupation, self-confidence, intervention, self-will, and perception were the most significant behavioural factors relating to hygiene behaviours. In addition, the study also confirmed that multilevel models are able to detect factors that influence variations at different levels. Furthermore, the study revealed that there were variations in hygiene behaviours among child caregivers at both the household and village levels.

## 6. Recommendations

Based on the results of this study, it is recommended that multilevel models be used when evaluating studies for interventions that are implemented in several households nested in different villages so as to better understand all of the risk factors that may be of importance in designing future interventions.

The results of this study provide information on factors that should be considered when designing hygiene-related interventions in rural areas so as to make sure there is a behavioural change among rural populations. It is therefore recommended that such factors be taken into consideration when implementing interventions aiming to change caregivers’ behaviours regarding sanitation in countries of similar context.

The contributions of this paper are two-fold. First, it contributes to research on the variation in hygiene behaviours among child caregivers at the household and village levels in southern Malawi by adopting a multilevel modelling approach. Second, it contributes to the understanding of risk factors within a district based on household and village characteristics.

## 7. The Strengths and Limitations of the Study

The strength of this study is that it used models that have been proven in the literature to detect risk factors associated with changes in behaviours among subjects. However, the main limitation of the study is that it used data that were collected without other valuable parameters that could have worked well with multilevel models. The researchers addressed this limitation by systematically transforming variables with three levels and more into binary variables to fit with the models.

This study relied on self-reported health behaviours, which are prone to bias [26]. However, this was controlled by conducting spot checks on the outcome variables (i.e., handwashing with soap, washing utensils with soap, and keeping utensils in a safe place) that were reported by the participants.

## Figures and Tables

**Figure 1 ijerph-22-00801-f001:**
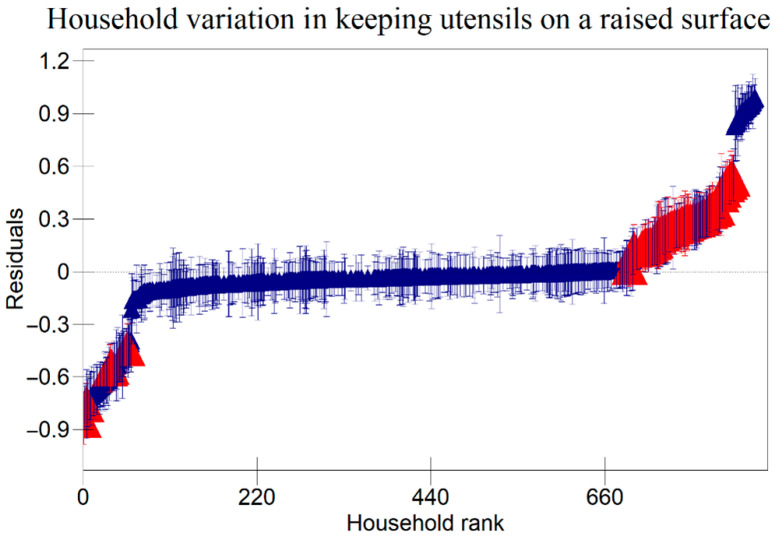
Variation in Keeping Kitchen Utensils on a Raised Surface Among Child Caregivers Between Households in Relation to the Area of Intervention and the Non-Intervention Area (A 95% credible interval caterpillar plot showing the residuals of keeping utensils on a raised surface ranked by their respective households). The red represents Households from non-intervention area and the blue represents Households from intervention area.

**Figure 2 ijerph-22-00801-f002:**
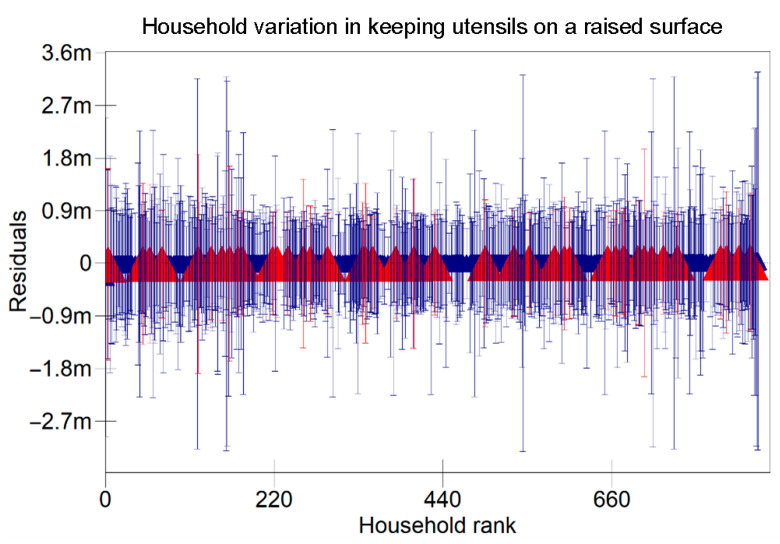
Variation in Using Soap to Clean Kitchen Utensils Among Child Caregivers Between Households in Relation to the Area of Intervention and the Non-Intervention Area (A 95% credible interval caterpillar plot showing the residuals of using soap to clean kitchen utensils ranked by their respective households). The red represents Households from non-intervention area and the blue represents Households from intervention area.

**Figure 3 ijerph-22-00801-f003:**
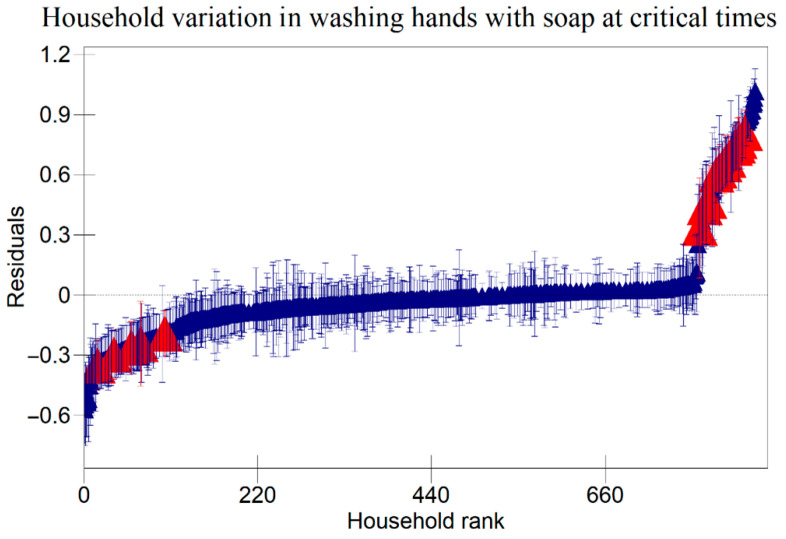
Variation in Washing Hands with Soap at Critical Times Among Child Caregivers Between Households in Relation to Area of Intervention and Non-Intervention Area (A 95% credible interval caterpillar plot showing the residuals of washing hands with soap at critical times ranked by their respective households). The red represents Households from non-intervention area and the blue represents Households from intervention area.

**Figure 4 ijerph-22-00801-f004:**
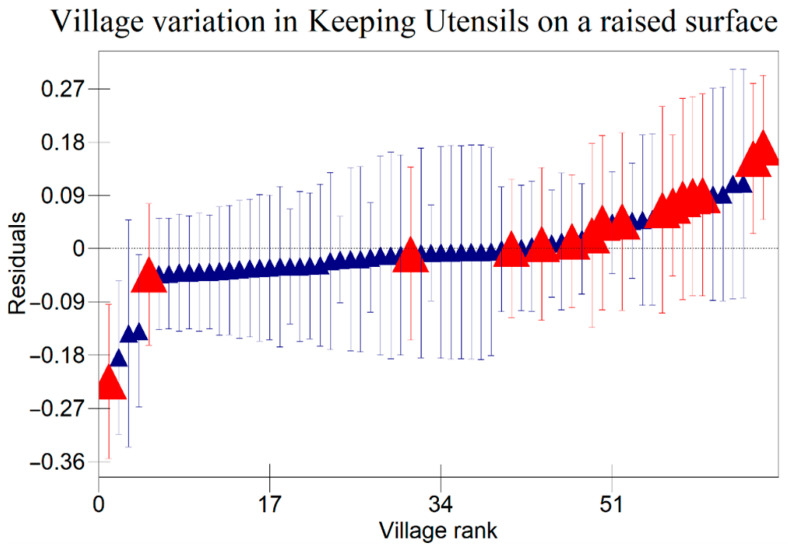
Variations in Keeping Kitchen Utensils on a Raised Surface Among Child Caregivers Between Villages in Relation to the Area of Intervention and the Non-Intervention Area (A 95% credible interval caterpillar plot showing the residuals of keeping utensils on a raised surface ranked by their respective villages). The red represents Villages from non-intervention area and the blue represents Villages from intervention area.

**Figure 5 ijerph-22-00801-f005:**
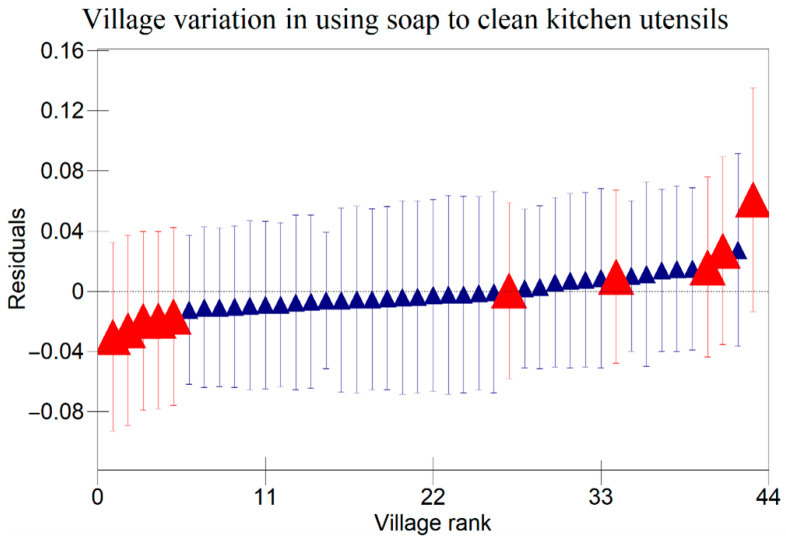
Variations in Using Soap to Clean Kitchen Utensils Among Child Caregivers Between Villages in Relation to Area of Intervention and Non-Intervention Area (A 95% credible interval caterpillar plot showing the residuals of using soap to clean kitchen utensils ranked by their respective villages). The red represents Villages from non-intervention area and the blue represents Village from intervention area.

**Figure 6 ijerph-22-00801-f006:**
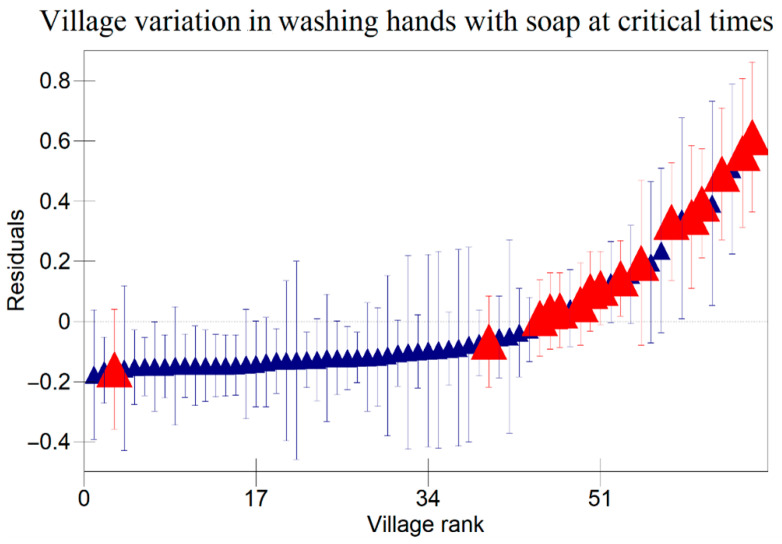
Variation in Washing Hands with Soap at Critical Times Among Child Caregivers Between Villages in Relation to Area of Intervention and Non-Intervention Area (A 95% credible interval caterpillar plot showing residuals of washing hands with soap at critical times ranked by their respective villages). The red represents Villages from non-intervention area and the blue represents Villages from intervention area.

**Table 1 ijerph-22-00801-t001:** Questions on the three targeted behaviours.

Behaviour	Items	Answer format
Hand washing before eating main meal	Before you feed your child main meals (e.g., lunch), how often do you wash your hands with soap and water?	(Almost) at no time–(almost) each time (5-point rating scale)
Hand washing after using toilet	After you defecate, how often do you wash your hands with soap and water?	(Almost) at no time–(almost) each time (5-point rating scale)
Hand washing before food preparation	Before you prepare food, how often do you wash your hands with soap and water?	(Almost) at no time–(almost) each time (5-point rating scale)
Hand washing before eating snacks	Before you feed your child snacks, how often do you wash your hands with soap and water?	(Almost) at no time–(almost) each time (5-point rating scale)
Hand washing after cleaning child’s bottom	After cleaning child’s bottom, how often do you wash your hands with soap and water?	(Almost) at no time–(almost) each time (5-point rating scale)
Washing kitchen utensils with soap	Before you use kitchen utensils, how often do you wash them with soap and water?	(Almost) at no time–(almost) each time (5-point rating scale)
Keeping utensils on an elevated place	Do you keep your kitchen utensils on an elevated place?	Not at all–very much (5-point rating scale)

Response scales: 5-point rating scale (ranging from “[almost] at no time” to “[almost] each time”; ranging from “not at all” to “very much”).

**Table 2 ijerph-22-00801-t002:** Descriptions of study’s explanatory variables.

Variable	Description	Measurement
Education	Highest education level of child caregiver	Categorical variable: 1 = never been to school, 2 = primary, 3 = secondary and above
Age group	Age group of child caregiver	Categorical variable: 1 = 18–28, 2 = 29–38, 3 = 39–48, 4 = 49–58, 5 = over 58
Marital status	Marital status of child caregiver	Binary variable: 0 = single, 1 = married
Occupation	Occupation of child caregiver	Categorical variable: 1 = farming, 2 = business, 3 = employed
Intervention	Location of child caregiver; whether they were in intervention area or not	Binary variable: 0 = not in intervention, 1 = in intervention
Relative wealth	Social-economic status of child caregiver	Categorical variable: 1 = high, 2 = middle, 3 = low
Health knowledge	Child caregiver’s knowledge of diarrhoea disease	Binary variable: 0 = not knowledgeable, 1 = knowledgeable
Risk	Child caregiver’s understanding of risk factors associated with diarrhoea	Binary variable: 0 = at risk, 1 = not at risk
Effort	Whether child caregiver found performing proper hygiene behaviour difficult or easy	Binary variable: 0 = difficult, 1 = easy
Norm	Whether child caregiver’s behaviours are results of guidance from local authorities	Binary variable: 0 = no, 1 = yes
Confidence	Confidence of child caregiver when performing proper hygiene behaviours	Binary variable: 0 = not confident, 1 = confident
Willingness	Willingness of child caregiver to perform proper hygiene behaviours	Binary variable: 0 = not willing, 1 = willing
Perception of price	Child caregiver’s perception of price of soap used for household sanitation use	Binary variable: 0 = not expensive, 1 = expensive

**Table 3 ijerph-22-00801-t003:** Demographic characteristics of study participants.

Variable	Description	Frequency (%)
Education	Never	21
	Primary	70
	Secondary and above	10
Age group	18–28	59
	29–38	29
	39–48	10
	49–58	1
	Over 58	1
Marital status	Single	9
	Married	91
Occupation	Farming	72.9
	Business	14.9
	Employed	12.1
Relative Wealth	Low	86.5
	Middle	10.8
	High	2.7

**Table 4 ijerph-22-00801-t004:** Descriptive estimates of non-doers of washing hands with soap at critical times, using soap to clean kitchen utensils, and keeping utensils on raised surface between area of intervention and non-intervention area at household level.

Behaviour	Category	Caregivers in Non-Intervention Area (%)	Caregivers in Intervention Area (%)
Keeping utensils on a raised surface	Keep utensils on a raised surface	32	96
	Do not keep utensils on a raised surface	68	4
Washing hands with soap at critical times	Wash hands with soap	70	97
	Do not wash hands with soap	30	3
Using soap to clean kitchen utensils	Use soap to clean kitchen utensils	78	97
	Do not use soap to clean kitchen utensils	22	3

**Table 5 ijerph-22-00801-t005:** Hierarchical binary logistic regression to identify food hygiene behavioural factors of household and village.

	Keep Utensils on a Raised Surface			Use Soap for Cleaning Utensils			Wash Hands with Soap at Critical Times		
Predictor	Β	t-Value	*p*-Value	Β	t-Value	*p*-Value	β	t-Value	*p*-Value
Educational Background									
Never	Reference group								
Primary	0.013	0.5	0.62	0	−0.09	0.93	0.043	1.87	0.62
Secondary and above	0.004	0.103	0.92	−0.03	−0.94	0.35	0.076	2.11	0.04 **
No									
Yes	0.019	0.358	0.72	−0.03	−0.56	0.58	0.062	1.29	0.2
Age Group									
18–28	Reference group								
29–38	−0.05	−2.22	0.03 **	−0.02	−0.79	0.43	0.031	1.48	0.14
39–48	−0.01	−0.29	0.77	0.011	0.367	0.71	−0.009	−0.28	0.78
49–58	−0.12	−1.23	0.22	0.036	0.439	0.66	0.142	1.65	0.99
Over 58	−0.07	−0.47	0.64	−0.05	−0.43	0.67	0.079	0.61	0.54
**Marital Status**									
Unmarried	Reference group								
Married	−0.01	−0.19	0.85	0.018	0.6	0.55	0.035	1.09	0.27
**Occupation**									
Farming	Reference group								
Business	−0.03	−1.07	0.28	0.008	0.333	0.74	−0.031	−1.19	0.23
Employed	0.036	1.161	0.25	0.073	2.808	**0.005 ****	−0.075	−2.68	**0.008 ****
**Intervention**									
No	Reference group								
Yes	−0.63	−24	**<0.00001 ****	−0.17	−7.91	**<0000.1 ****	0.256	11.1	**<0.00001 ****
**Relative Wealth**									
Low	Reference group								
Middle	0.077	1.75	0.08 *	−0.01	−310	0.76	−0.034	−1.06	0.28
High	−0.04	−0.5	0.62	−0.07	−1.33	0.1 *	0.086	1.43	0.1 *
**Health Knowledge**									
Not knowledgeable	Reference group								
Knowledgeable	0.004	0.2	0.84	−0.01	−0.29	0.77	−0.012	−0.67	0.5
**Risk**									
Not at risk	Reference group								
At risk	0.098	1.4	0.16	0.068	1.133	0.26	0.012	0.19	0.85
**Effort**									
Difficult	Reference group								
Easy	0.008	0.195	0.85	0.047	1.343	0.18	−0.018	−0.49	0.63
**Norm**									
No	Reference group								
Yes	0	−0.07	0.95	−0.01	−0.03	0.97	−0.008	−0.2	0.85
**Confidence**									
Not confident	Reference group								
Confident	−0.3	−3.43	**0.0006 ****	−0.09	−1.24	0.22	0.224	2.84	**0.005 ****
**Willingness**									
Not willing	Reference group								
Willing	0.001	0.008	0.994	0.072	0.692	0.49	−0.201	−1.83	0.07 *
**Perception of Price**									
Not expensive	Reference group								
Expensive	0.013	0.5	0.62	0.102	4.636	**<0.00001 ****	−0.098	−0.43	0.67
**Household Effects ** $\left(e_{0ij}\right)$	0.003	1.5	**<0.00001 ****	0.001	1	**<0.00001 ****	0.001	1	**<0.00001* ***
**Village Effects** $\left(u_{0j}\right)$	0.076	19	**<0.00001 ****	0.056	18.67	**<0.00001 ****	0.063	21	**<0.00001 ****
**DIC**	**216.59**			**−34.13**			**66.71**		

* = marginally significant; ** = highly significant.

## Data Availability

The raw data supporting the conclusions of this article will be made available by the authors upon request.

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
