# Peer review of "Exploring Hygiene Behaviours Among Child Caregivers in Rural Malawi Using Multilevel Logistic Models"

_ijerph, 2025, doi:10.3390/ijerph22050801_

Round 1
Reviewer 1 Report
Comments and Suggestions for Authors
Measurement of handwashing
No information about how handwashing was measured. It is highly inaccurate to measure handwashing through self-report on a questionnaire only. This is demonstrated in many articles, such as this early study in DR Congo:
Manun'Ebo M, Cousens S, Haggerty P, Kalengaie M, Ashworth A, Kirkwood B. Measuring hygiene practices: a comparison of questionnaires with direct observations in rural Zaïre. Trop Med Int Health. 1997 Nov;2(11):1015-21. doi: 10.1046/j.1365-3156.1997.d01-180.x.
I looked at reference 19: Chidziwisano K, Slekiene J, Kumwenda S, Mosler HJ, Morse T. Toward Complementary Food Hygiene Practices among Child Caregivers in Rural Malawi. Am J Trop Med Hyg. 2019 Aug;101(2):294-303. doi: 10.4269/ajtmh.18-0639.
It mentions structured observation, spot checks, and questions about handwashing at different key times on a household survey.
There is no specific information provided about the origin of the handwashing variables, that are the basis of all subsequent analysis. Without this information, it is not possible to assess the appropriateness of the regression analysis and other analyses
Comments on the Quality of English LanguageGrammatical problems. There are multiple grammatical problems throughout the manuscript. It is not a good use of my time as a reviewer to be assigned a manuscript with multiple grammatical and proofreading problems. Therefore I have paused my review.
1. Line 12 - It was done so by exploring factors relating to three hygiene behaviours among child caregivers. (Odd impersonal construction)
2. Line 15 -Through this research, it was as well aimed to explore variations at household and village levels by the use of multilevel binary logic 16 models. (Odd impersonal construction)
3. Lines 77-81 - The study mainly focused on variations between area of intervention as a treatment and area where intervention was not carried out as a control. The project delivered various interventions which were in categories of treatments in respective villages. However, this study focussed on the overall impact of all the interventions as they were all delivered for the same purpose of trying to change caregivers’ hygiene behaviours. (Overuse of the verb focus. Two different spellings: focused and focussed).
4. Lines 83-87 - And by looking at that variable in the regression model, it was able to tell whether there were differences between the control and the treatment. It was done so by assessing variation amongst child caregivers in relation to three hygiene behaviours; washing hands with soap at critical times, storing kitchen utensils on a raised surface and using soap to clean kitchen utensils. (Off impersonal constructions)
5. Lines 115-117 - Incorrect numbering of aims
6. Lines 134-135 - Descriptions of explanatory variables included in the model are shown in the Error! Reference source not found..
7. Line 250 - , the results indicate that there were variations at household level on keeping utensils 250 on an elevated surface, using soap to clean kitchen utensils, and washing hands with soap 251 at critical times. (Sentence starts with a comma)
Author Response
Thank you very much for taking the time to review this manuscript. Please find the attached table for your guidance to changes made highlighted in blue.

Reviewer 2 Report
Comments and Suggestions for Authors
Major comments:
1. Title
Suitability of Title
Your title “Exploring Hygiene Behaviours amongst Child Caregivers in Rural Malawi Using Multilevel Logistic Models” is clear and descriptive, which is great! It effectively conveys the focus of your research on hygiene behaviors, the specific population (child caregivers), the location (rural Malawi), and the methodological approach (multilevel logistic models). Here is a slight tweak to improve readability: I changed "amongst" to "among" for simplicity and consistency in American English. If you prefer British English, "amongst" is perfectly fine. Overall, it looks suitable and informative!
2. Abstract
Old Abstract version: This study aimed to explore factors that influence food hygiene behaviours among child caregivers in Chikwawa district, Malawi. It was done so by exploring factors relating to three hygiene behaviours among child caregivers: keeping utensils on an elevated surface, using soap to clean kitchen utensils, and washing hands with soap at critical times. Research has shown that these behaviours contribute to the reduction of diarrheal disease. Through this research, it was as well aimed to explore variations at household and village levels by the use of multilevel binary logic models. The results have revealed that education background, age group, occupation, self-confidence, intervention, self-will, and perception were the most significant behavioural factors relating to food hygiene behaviours. There were variations at village level (p<0.00001) and there were no variations at household level (p>0.1). It has also revealed that Caregivers from the area of intervention had a positive response to the interventions.
New Abstract version: This is a well-structured abstract for your study. It clearly outlines the aim, methods, and key findings. If you want to refine it further, consider the following enhanced version for clarity and flow:
This study aimed to explore the factors influencing food hygiene behaviors among child caregivers in the Chikwawa district, Malawi. The research focused on three specific hygiene behaviors: keeping utensils on an elevated surface, using soap to clean kitchen utensils, and washing hands with soap at critical times. These practices are known to contribute to the reduction of diarrheal disease. To understand these behaviors, the study utilized multilevel binary logistic models to examine variations at both household and village levels. The findings revealed that educational background, age group, occupation, self-confidence, intervention, self-will, and perception were the most significant factors influencing food hygiene behaviors. Notably, there were significant variations at the village level (p<0.00001), while no significant variations were observed at the household level (p>0.1). Additionally, caregivers from areas where interventions were implemented showed a positive response to these interventions.
Double abstract
The abstract was repeated. Kindly delete one from the main manuscript [Line 11 to Line 36].
3. Introduction
A research introduction is the opening section of a research paper, and it sets the stage for the entire study. The introduction provides context by presenting background information on the topic. It helps the reader understand the broader field of study and why the specific research question is important. This section clearly outlines the problem or issue that the research aims to address. It explains why this problem is significant and warrants investigation. The introduction details the specific objectives of the study and the research questions or hypotheses that will guide the investigation. Here, the researcher explains the importance of the study and its potential contributions to the field. It may also discuss any gaps in existing research that the study aims to fill. The introduction often outlines the scope of the research, indicating what will and will not be covered. It may also acknowledge any limitations or constraints that could impact the study. A brief overview of the research methodology is provided, giving the reader an idea of how the study will be conducted. Thus, a well-crafted introduction engages the reader, sets up the research question, and provides a clear roadmap for the rest of the study. Hence, it should be concise, informative, and compelling. Therefore, the introduction should the ‘funnel’ shape approach review and the questions below can help fathom the introduction better. Look at the questions below:
How do cultural norms and practices in rural Malawi influence hygiene behaviors among caregivers? Are there specific local customs that either support or hinder these practices?
What types of interventions have been most effective in changing hygiene behaviors in similar settings? Are there any innovative approaches that could be tested in this context?
How do hygiene behaviors in Chikwawa district compare with those in other regions of Malawi or similar settings in other countries? Are there lessons to be learned from these comparisons?
How does the socioeconomic status of caregivers influence their hygiene behaviors? Are there specific economic barriers that prevent them from adopting better hygiene practices?
What is the rationale for this study? Kindly include it, as this will help to understand the study significance.
4. Theoretical and Conceptual framework
Can you include any suitable behavioral theory(ies) that can be applied and use to understand the factors influencing hygiene behaviors? For instance, utilizing Theory of Planned Behavior or Social Cognitive Theory to explain this study and the implications of your findings.
5. Methodology
The numbering in the outcome variable should be checked [Line 113 – Line 117]
Check Line 134 to 135 and address the errors there
Address Line 158 to Line 160 by using the Mathematical equation editor on Microsoft Word
Check Line 190 and address the error
Determination of sampling size – here you will describe how the sampling size of 569 was gotten. Discuss the formula you used in deriving the figure of 569. Also, how did you derive at 626 respondents with a 10% of non-response rate? What is your justification for the 10% of non-response rate? Kindly address this.
Variable measurements (Outcome variable or Dependent variable) – What is the dependent variable of this study? How do derive the dependent variable? Which of the study research questions did you use in measuring the dependent variable? Insert the questions from the research questionnaire instruments. Note that a dependent variable is the variable that changes as a result of the independent variable (independent or explanatory variables or factors) manipulation. It's the outcome you're interested in measuring, and it “depends” on your independent variable. In statistical analysis, dependent variables are also called ‘response variables’ (they respond to a change in another variable). Therefore, specify your outcome variable very well and show us how it is going to be measured. Is it a binary outcome variable or what? This section should discuss the measurements of the outcome variable or dependent variable(s). Kindly address this.
Explanatory Variable or Independent Variable: You did not indicate your independent variables you used in this study. I know that you cannot have the dependent variable without the independent variable when running the statistical analysis. Insert the independent variables and how they are going to be measured (References may be cited). Define all your Explanatory Variables or Independent Variables and how they were measured in your study. This is very important, as it will affect the process and approach of statistical analysis. Kindly address this.
How can the effectiveness of hygiene interventions be better measured and evaluated? Are there new tools or methodologies that could enhance the accuracy of your findings?
Also, the definition of concepts used in this study will come after 10b. Here, you will give a brief explanation of the concepts and how these concepts were operationalized in your study. Kindly address this.
6. Results
Prepare a Table for Line 173 to Line 183
Line 112 to Line 132 if you have 3 levels of outcome variable using 5 Likert scale of measurements, you should specify the appropriate type of analysis you want to use. The choice of analysis you used in your study is not appropriate. For instance, when you have an outcome variable with three levels, the type of analysis you choose will depend on the nature of your data and your research objectives. Here are a few possible options:
Multinomial Logistic Regression: This is a commonly used method when your dependent variable is categorical with more than two levels. It allows you to model the probability of each outcome category as a function of independent variables.
Ordinal Logistic Regression: If the three-level outcome variable has a natural order (e.g., low, medium, high), ordinal logistic regression might be appropriate. This method takes the ordering into account when modeling the relationship between the outcome and predictor variables.
ANOVA (Analysis of Variance): If the independent variables are categorical and you want to compare the means of a continuous variable across the three levels of your outcome variable, ANOVA can be used.
Discriminant Analysis: This method is useful for identifying which variables discriminate between the three groups. It can be particularly helpful when you aim to predict group membership based on a set of predictors.
The choice of method will depend on the specific characteristics of your data and your research questions. From Line 155, you said you use the binary regression model and binary regression model, also known as logistic regression, is used when you have a binary dependent variable—that is, an outcome variable with two possible values, often coded as 0 and 1. You have to address the analysis section.
Kindly report the results according to the objectives using the objective as a theme heading. Show the type of analysis employed and elaborate on the details of the study findings. Therefore, kindly revise. The results should include all the interpretations for the Tables, and the Table legend should be carefully and appropriately inserted in the primary research. Interpret and discuss the relevant and significant variables and make them precise so they don’t lose meaning. Your readers will see the tables with the results. Kindly address these concerns mentioned.
7. Discussion
The discussion section is one of the final parts of a research paper, in which an author describes, analyzes, and interprets their findings. They explain the significance of those results and tie everything back to the research question(s). It should focus on explaining and evaluating what you found, showing how it relates to your literature review and paper or dissertation topic, and making an argument supporting your overall conclusion. Therefore, let your discussion aspects focus on interpreting your findings and relating them to literature, whether they corroborate or not, with existing studies, alongside the in-citation of recent references. Let each paragraph relate to the objective findings. Discuss in assertion or not with the study’s findings from other studies. However, there are few questions that will help to build the discussion section and tighten the aspects that were not included in the discussion section. These questions are as follows:
How do gender roles within households affect hygiene practices? Are there differences in behavior between male and female caregivers, and how can interventions be tailored accordingly?
What is the impact of environmental factors, such as water quality and access to sanitation facilities, on hygiene behaviors? How do these factors interact with the individual-level variables you have identified?
What is the current level of health education among child caregivers in rural Malawi? How effective are existing health education programs, and what improvements can be made?
How involved are local communities in promoting hygiene behaviors? Can community leaders or local organizations play a more significant role in intervention efforts?
Are there cultural beliefs or practices that impact hygiene behaviors? How can interventions be designed to be culturally sensitive and respectful?
What is the availability of essential resources like clean water, soap, and hygiene products in the study area? How does access (or lack thereof) affect hygiene behaviors?
How does the availability and condition of infrastructure, such as latrines and handwashing stations, influence hygiene behaviors?
What are the psychological or behavioral barriers that prevent caregivers from adopting better hygiene practices? How can these barriers be addressed?
What factors contribute to the long-term sustainability of hygiene interventions? How can programs ensure that behavior changes are maintained over time?
Addressing those questions raised above in the discussion section will help a lot. Kindly revise.
8. Strengths and Limitations of this Study?
Kindly explain in detail the strengths and limitations of this study. Indicate the Potential bias encountered in this study.
9. Implications of findings for future studies
What is the contribution of this study to the existing one, especially in sub-Saharan African or developing countries? What are the implications of your findings across other countries with lower prevalence of your outcome of interest?
10. Conclusion
The conclusion should stem from summing up your research paper (by the following steps - restate your research topic, restate the thesis, summarize the main points, state the significance or results, and conclude your thoughts). Remember that a conclusion is not merely a summary of the main topic(s) covered or a re-statement of your research problem but a synthesis of key points and, if applicable, where you should recommend new areas for future research.
11. Recommendations
Where is the Recommendation for this study? You can add it to the Conclusion section with the subheading – Conclusion/ Recommendations.
12. Language Editing
Ask for a Professional English editor to address all grammatical errors and lexis structure of your manuscript.
13. References
Kindly check your references and see if they are fully citied in the main content of your research. Kindly check and revisit them all.
Suggested Readings
Chidziwisano, K., Slekiene, J., Kumwenda, S., Mosler, J., & Morse, T. (2019). Toward Complementary Food Hygiene Practices among Child Caregivers in Rural Malawi. The American Journal of Tropical Medicine and Hygiene, 101(2), 294. https://doi.org/10.4269/ajtmh.18-0639
Ogutu, E.A., Ellis, A., Rodriguez, K.C. et al. Determinants of food preparation and hygiene practices among caregivers of children under two in Western Kenya: a formative research study. BMC Public Health 22, 1865 (2022). https://doi.org/10.1186/s12889-022-14259-6
Bozicevic, L., Lucas, C., Magai, D. N., Ooi, Y., Maliwichi, L., Sharp, H., & Gladstone, M. (2024). Evaluating caregiver-child interactions in low- and middle-income countries: a systematic review of tools and methods. Journal of Reproductive and Infant Psychology, 1–36. https://doi.org/10.1080/02646838.2024.2321615
Chidziwisano, K., Tilley, E., & Morse, T. (2020). Self-Reported Versus Observed Measures: Validation of Child Caregiver Food Hygiene Practices in Rural Malawi. International Journal of Environmental Research and Public Health, 17(12), 4498. https://doi.org/10.3390/ijerph17124498
Watson J, Okumu NO, Wasonga JO, Majiwa H, Kiarie A, Masudi SP, et al. (2024) A proof-of-concept randomised controlled trial of an intervention designed to improve food hygiene behaviours among caregivers of young children living in low-income areas of Nairobi, Kenya. PLOS Water 3(7): e0000223. https://doi.org/10.1371/journal.pwat.0000223
L. Bozicevic, C. Lucas, D. N. Magai, Y. Ooi, L. Maliwichi, H. Sharp & M.Gladstone (05 Mar 2024): Evaluating caregiver-child interactions in low- and middle-income countries: a systematic review of tools and methods, Journal of Reproductive and Infant Psychology, DOI: 10.1080/02646838.2024.2321615

Ask for a Professional English editor to address all grammatical errors and lexis structure of your manuscript.
Author Response

(The authors gave the same response as above.)

Round 2
Reviewer 1 Report
Comments and Suggestions for Authors_
Reviewer 2 Report
Comments and Suggestions for Authors
The authors have fully addressed the comments raised by all Reviewers.
Comments on the Quality of English LanguageThe quality of the English Language is good.